# Surface Pretreatments of AA5083 Aluminum Alloy with Enhanced Corrosion Protection for Cerium-Based Conversion Coatings Application: Combined Experimental and Computational Analysis

**DOI:** 10.3390/molecules26247413

**Published:** 2021-12-07

**Authors:** Mohammad Reza Shishesaz, Moslem Ghobadi, Najmeh Asadi, Alireza Zarezadeh, Ehsan Saebnoori, Hamed Amraei, Jan Schubert, Ondrej Chocholaty

**Affiliations:** 1Department of Technical Inspection Engineering, Abadan Faculty of Petroleum Engineering, Petroleum University of Technology, Abadan P.O. Box 63187-14317, Iran; amraehamed69@gmail.com; 2School of Metallurgy and Materials Engineering, College of Engineering, University of Tehran, Tehran P.O. Box 11155-4563, Iran; ghobadi.put92@gmail.com (M.G.); najmehasadi@ut.ac.ir (N.A.); 3Department of Safety Engineering, Abadan Faculty of Petroleum Engineering, Petroleum University of Technology, Abadan P.O. Box 63187-14317, Iran; alireza.zarezadeh97@gmail.com; 4Advanced Materials Research Center, Department of Materials Engineering, Najafabad Branch, Islamic Azad University, Najafabad P.O. Box 15847-43311, Iran; saebnoori@pmt.iaun.ac.ir; 5Department of Material and Metallurgy, University of West Bohemia, Univerzitní 22, 306 14 Pilsen, Czech Republic; schubert@vzuplzen.cz (J.S.); chochola@kmm.zcu.cz (O.C.)

**Keywords:** surface pretreatment, cerium conversion coatings, artificial neural network, modeling, anfis

## Abstract

The effects of surface pretreatments on the cerium-based conversion coating applied on an AA5083 aluminum alloy were investigated using a combination of scanning electron microscopy (SEM), energy-dispersive X-ray spectroscopy (EDS), polarization testing, and electrochemical impedance spectroscopy. Two steps of pretreatments containing acidic or alkaline solutions were applied to the surface to study the effects of surface pretreatments. Among the pretreated samples, the sample prepared by the pretreatment of the alkaline solution then acid washing presented higher corrosion protection (~3 orders of magnitude higher than the sample without pretreatment). This pretreatment provided a more active surface for the deposition of the cerium layer and provided a more suitable substrate for film formation, and made a more uniform film. The surface morphology of samples confirmed that the best surface coverage was presented by alkaline solution then acid washing pretreatment. The presence of cerium in the (EDS) analysis demonstrated that pretreatment with the alkaline solution then acid washing resulted in a higher deposition of the cerium layer on the aluminum surface. After selecting the best surface pretreatment, various deposition times of cerium baths were investigated. The best deposition time was achieved at 10 min, and after this critical time, a cracked film formed on the surface that could not be protective. The corrosion resistance of cerium-based conversion coatings obtained by electrochemical tests were used for training three computational techniques (artificial neural network (ANN), adaptive neuro-fuzzy inference system (ANFIS), and support vector machine regression (SVMR)) based on Pretreatment-1 (acidic or alkaline cleaning: pH (1)), Pretreatment-2 (acidic or alkaline cleaning: pH (2)), and deposition time in the cerium bath as an input. Various statistical criteria showed that the ANFIS model (*R*^2^ = 0.99, MSE = 48.83, and MAE = 3.49) could forecast the corrosion behavior of a cerium-based conversion coating more accurately than other models. Finally, due to the robust performance of ANFIS in modeling, the effect of each parameter was studied.

## 1. Introduction

Lightweight materials such as aluminum alloys are attractive for applications in many different industries, including automobile, aviation, aerospace, biological implants, and sports equipment [1]. The application of aluminum alloys is limited due to lower electrochemical stability and thus poor corrosion properties, especially in brine environments. The halide ions, especially Cl- ions, are very harmful to the aluminum oxide layer. They adsorb on the surface, react with aluminum and generate a defective, weaker, thinner oxide layer, which causes pitting corrosion [2]. In fact, all aluminum alloys are covered by a natural oxide layer protecting the alloy against further dissolution.

Many strategies have been used as protective methods for aluminum alloys. Although chromate conversion coatings (CrCCs) have been extensively used to achieve high corrosion resistance [3], Cr is toxic and carcinogenic [4]. Thus, a variety of environmentally friendly conversion coatings have been developed, such as phosphate conversion coatings [5], molybdate conversion coatings [6], zirconium/titanium conversion coatings [7,8], and cerium conversion coatings (CeCCs) [9,10]. These coatings can improve Al alloys’ corrosion resistance and adhesion properties, making them promising alternatives to Cr conversion coating [11].

The cerium compounds deposit on the aluminum surface and form a protective oxide layer to decrease the corrosion rate of aluminum alloys [12,13]. According to the related literature, CeCCs have been previously investigated on aluminum alloys, and suitable corrosion inhibition behaviors have been observed, limiting the access of water and corrosive ions to the substrate [14,15]. Several factors affected the microstructure and the corrosion resistance of conversion coatings [16,17]. Experimental parameters such as surface pretreatment should be controlled to improve the coating properties. It has been reported that surface pretreatment helps to increase the corrosion resistance of CeCCs [18]. The changes in temperatures used for alkaline cleaning affect the coating thickness and corrosion resistance [19]. Moreover, previous research has shown that pretreatment with a 10-wt.% sulfuric acid solution can increase the deposition rate significantly [20]. B. Valdez et al. [21] investigated the corrosion behavior of CeCCs on Al-6061 alloy, the effect of hydrogen peroxide and pH of the bath was discussed. Brunelli et al. [10] reported that the adhesion and corrosion resistance of CeCCs on magnesium alloy would be increased by performing HCl pretreatment. Maddela et al. [22] evaluated the influence of sulfuric acid and sodium carbonate surface pretreatments on the corrosion behavior of CeCCs applied on AZ91D magnesium alloy. It was understood that the morphology and crack density of CeCCs were affected by surface pretreatments. In addition, acid and alkaline pretreatments together resulted in better corrosion resistance than individual surface pretreatment. In other research, the effect of surface pretreatment in the deposition of CeCCs on 7075-T6 and 2024-T3 alloys was studied. The results demonstrated that surface pretreatment significantly improves the morphology of the coatings [23].

AA5083 contains more magnesium than the other alloying elements. This alloy consists of (Al-Fe-Mn rich) particles and Mg-Si rich particles, dispersed in the matrix [24,25,26,27,28,29,30]. Corrosion of AA5083 in aggressive NaCl solutions should be studied from several points of view. Cl^−^ in NaCl solution destroys the natural oxide layer formed on the matrix AA5083 [31,32]. Further, the alloy is affected by the localized attack in the surrounding intermetallic compounds’ presence in the matrix. The main types of intermetallic inclusions in the 5083 alloy are iron-rich and magnesium-rich intermetallics [26].

The magnesium-rich intermetallics have an anodic behavior and demonstrate partial dissolution with distinct dealloying due to the selective leaching of magnesium. The formation of hydroxide deposits and the enrichment of the intermetallics in silicon stop further propagation of defects, preventing deep pits’ formation. The iron-rich intermetallic phase has more noble potential relative to the aluminum matrix. This potential difference is because of the formation of galvanic couples that leads to the localized attack near the Fe-rich phase. The intermetallic inclusion works as a cathode promoting fast anodic dissolution of aluminum and formation of the pit [27,28].

In fact, the presence of intermetallics (Fe rich) act as a cathodic site suitable for the reduction of O_2_ and generation of OH^−^ according to the following reaction:(1)$$O_{2}+2\;H_{2}O\to 4\;{\mathrm{OH}}^{-}$$

This reaction causes an increase in pH around these cathodic sites and dissolves the natural aluminum oxide around these particles. Therefore, cathodic intermetallic particles provoke alkaline localized corrosion in the metallic matrix surrounding the precipitates [33]. The main concern is the presence of cathodic intermetallic compounds causing localized alkaline corrosion of AA5083 [31].

Hasannejad et al. [34], applied a CeCC on AA5083. Their results showed that application of CeCC on the AA5083, led to shift of the polarization curves to left and also Ecorr shifted to more negative values. These results state that cerium oxide behaves as cathodic inhibitor for aluminum alloys. They claimed that by increasing time of immersion in Ce containing solution up to 30 min, Icorr of coated samples decreased but after 30 min the parameter increased. This means there is an optimum time for dipping of alloy in ce containing solution as after that microcracks may appeared.

Dabala et al. [35], assessed the influence of several parameters, such as temperature, dipping time, concentration of cerium ions and H_2_O_2_, pH of the conversion solution, on the composition and morphology of the CeCC as well as corrosion resistance of AA5083.

Nowadays, many researchers utilize intelligent reasoning systems to significantly improve product quality because of their capability in prioritization, optimization, planning, and forecasting [36]. Application of soft computing techniques such as artificial neural networks (ANN) and adaptive neuro-fuzzy inference systems (ANFIS), and support vector machine regression (SVMR) have recently garnered considerable attention because of their vast capabilities and flexibility of use as compared to other traditional modeling methods [37,38,39]. ANN is a well-known mathematical simulator that is inspired by the structure of the human brain. They have been ordered to have a capacity to accomplish similar to a human, by instructive data and learning activities [40]. It can learn the linear and nonlinear relevance between different variables from a dataset. In addition, this method can simulate various processes without the full realization of mathematical equations and can handle complicated engineering problems [41].

ANFIS is a hybrid universal tool that was first introduced by Jang in 1993 [42]. The capabilities of fuzzy logic systems were combined with ANN learning abilities in the ANFIS algorithm [43]. ANFIS assisted in modeling the experimental datasets by converting logical statements to mathematical relations [44]. Ghobadi et al. [44] utilized ANFIS and ANN modeling to study the corrosion resistance of Lanolin coatings obtained by impedance test at various immersion times. Bucolo et al. [45] introduced a novel neuro-fuzzy model to forecast the corrosion phenomena in a pulp and paper plant. Mousavifard et al. [46] also used the ANFIS system to predict the corrosion rate of the zirconium-based nano-ceramic layer on galvanized steel. Their results indicated the efficacy of computational models in predicting corrosion rate and the importance of input factors in the prediction process.

As statistical approaches, regression methods have also attracted researchers’ attention for improving artificial intelligence techniques in predicting the experimental data. For this purpose, SVMR has been presented as the effective method for performing regression [47]. This method has a very stable and robust algorithm with a high capability for simulating nonlinear relationships. Moreover, it is usually comparable to neural techniques and can predict the experimental variable [48]. As mentioned in the literature [14,15,16,17,18,19,20,21,22,23,49], Ce^3+^ ion has been shown to be very effective as a cathodic inhibitor for an aluminum metallic surface. Due to the presence of some intermetallics in AA 5083, a non-uniform film of natural oxide forms on the surface which are more noble or more active than the matrix, so corrosion pits could be created on the surface. Therefore, cerium probably could be able to cover cathodic intermetallic sites of AA5083. This study evaluates the corrosion resistance of cerium-based conversion coating on aluminum alloy 5083, prepared at acid and alkaline surface pretreatment conditions. Applying acidic or alkaline surface pretreatment removes the non-uniform oxide layer and makes the surface ready for more effective CeCC deposition. The elemental and chemical composition and surface morphology are characterized using SEM and EDS. The best processing conditions to obtain the most uniform and crack-free films are introduced using the characterization results. The corrosion resistance properties of cerium coatings are investigated using EIS and Tafel polarization tests. In addition, based on the dataset of inhibitory power obtained by the experimental test results, three soft computing models were adopted to build a model for simulating the corrosion resistance of coated Al-alloys at various pretreatment conditions.

## 2. Experimental Procedures

### 2.1. Sample Pretreatment and Preparation

Aluminum alloy samples of AA5083 with the chemical composition of (Si:0.4, Fe:0.4, Cu:0.1, Mn:0.5, Mg:4.5, Cr:0.15, Zn:0.25, Ti:0.15 and balance Al in wt. %) were used as a metallic substrate to study the effect of pretreatment on corrosion behavior and cerium coating quality deposited on the alloy surfaces. Materials used in this study were Merck products of analytical grade and were used without further purifications. All samples were cold mounted in epoxy resin to obtain an exposed area of 1cm2. The specimens then were subjected to mechanical polishing using silicon carbide papers of 400 to 2000 grade. The samples were degreased with acetone, and then various operations of surface preparation (pretreatment) on different aluminum pieces were applied, as presented in Table 1. The pH of the acidic and alkaline solution was 3 and 12 respectively. Between each step of the surface pretreatment process, the samples were rinsed with deionized water.

The protective layer was applied to different surface pretreatment conditions by immersing them in a cerium bath for 10 min. The coating bath consisted of 1.5 g/100 mL CeCl_3_.7H_2_O, some drops of glycerin (=glycerol) as an organic plasticizer, and 2 mL H_2_O_2_. According to literature [50], H_2_O_2_ helps to increase the deposition rate of reaction. After selecting the best surface pretreatment, various deposition times were investigated (1, 5, 10, and 20 min) to find the suitable deposition time in the cerium bath.

### 2.2. Electrochemical Tests

The potentiodynamic polarization (Tafel analysis) and electrochemical impedance spectroscopy (EIS) techniques were adopted to evaluate the corrosion resistance of the coatings. The electrochemical tests were performed using Potentiostat/Galvanostat Autolab model PGSTAT 302N with a flat electrochemical cell containing three entrances for a conventional three-electrode assembly where the coating material, saturated calomel electrode (SCE), and pure Pt rod were employed as the working, reference, and counter electrodes respectively. The tests were conducted in a 3.5 wt% NaCl solution at ambient temperature. The electrochemical measurement tests were arranged in a non-stirred state while the exposed area of all materials was fixed to 1 cm^2^. The frequency ranges from 100 kHz to 10 mHz with imposed AC amplitude of 10 mV was utilized for EIS experiments. In this test, the extraction process of the EIS data was always initiated after the OCP became stable. For Tafel analysis, the test was performed in the potential range of ± 300 mV around OCP with a scan rate of 1 mV/s. Whole experiments were conducted at least three times to ensure the reproducibility of the extracted data and provide a precise indication of the corrosion rate. Zview3.1 software was employed to fit the EIS results.

### 2.3. Surface Characterization

Finally, the morphology of the surface and chemical composition of the coatings were studied by scanning electron microscopy (SEM, VEGA, TESCAN-LMU, Brno, Czech Republic) equipped with an energy-dispersive X-ray spectroscopy (EDS) probe.

## 3. The Procedures for Computational Analysis

The influence of different surface pretreatments on the corrosion behavior of CeCCs was evaluated using three computational methods. The pH of surface pretreatments and deposition time of CeCCs were selected as inputs, and the coating’s resistance was set as the output for each model to construct each model. Therefore, three models were trained, and the accuracy of each model was examined through various error criteria. Finally, the most suitable and accurate model was proposed for modeling the corrosion behavior of CeCCs. The accuracy of the models was measured by several statistical errors, such as MAE, which was calculated according to Equation (2):(2)$$\mathrm{MAE}=\frac{1}{n}\sum_{i=1}^{n}\left|t-o\right|$$
and for calculating the mean-square error (MSE), the following relation can be used:(3)$$\mathrm{MSE}=\frac{1}{n}\sum_{i=1}^{n}{\left(t-o\right)}^{2}$$

In these equations, *t* is experimental data, *n* is the number of data for training, and *o* is predicted data. In addition, the coefficient of determination (*R*^2^) was calculated according to Equation (4):(4)$$R^{2}=\;1-\frac{\sum_{i=1}^{n-1}{\left(t-o\right)}^{2}}{\sum_{i=1}^{n-1}{\left(t-m\right)}^{2}}$$
where *m* represents the average of the test dataset values, the model has a suitable accuracy if a model results in a lower value of MAE and RMSE. In the case of the *R*^2^ value, the model which results closer to unity represents a better ability for modeling [51].

### 3.1. ANN Modeling

ANN is a computational approach that includes various biological neural structures. This computational method can achieve the appropriate relationships among input and output variables without any prerequisites. The three-layer fundamental system of neural networks comprises input, hidden, and output layers [52]. Accordant with the dataset, Pretreatment-1 (acidic or alkaline cleaning: pH (1)), Pretreatment-2 (acidic or alkaline cleaning: pH (2)), and deposition time in the cerium bath were set as the inputs for each model, and the charge transfer resistance of the coatings was selected as the output. According to Table 1, when Step 1 or Step 2 was not applied, the pH value for the step was considered as neutral (pH = 7).

The structure of ANN modeling employed in this study is shown in Figure 1. Each layer contains the basic factor neurons, which are processing factors. The neuron calculates the weight of each input with a particular weight index (w) received from the signal of an output. The total weights of the inputs show the bias (b) and the transfer function f (Σwixi). Thus, the neurons in different layers are connected. Finally, nonlinear mapping joins the input layer to the output layer. A transfer function or activation function turns the signal data that moves between each neuron. One of the crucial parts in the accuracy of the constructed model is the arrangement of hidden layers. The learning process and the quantity of each independent parameter were obtained [53].

The performance and convergence of the ANN model are highly affected by the number of hidden layers and neurons in each hidden layer. Even though, for a small dataset, a single hidden layer usually results in more accuracy and convergence than two hidden layers. However, these two elements have an essential effect on the efficiency of an ANN model. It was preferred to search for the best ANN architecture, such as a model with an optimizing training algorithm and the number of neurons in each hidden layer [54].

Thus, we seek to produce an appropriate structure with one hidden layer for the ANN model. For training, the Levenberg–Marquardt backpropagation (LMBP) algorithm was employed. LMBP has been widely used in prior research and is well-known for its speed, processing power, reliability, and simplicity [39]. Therefore, for changing the number of neurons, the LMBP training algorithm was utilized. The ANN modeling used hyperbolic tangent sigmoid and Purlin as transfer functions for the hidden and output layers. Several studies showed that fewer neurons resulted in underfitting, and more neurons could result in overfitting, so trial and error are required to extend the trustable structure for ANN [55]. In this study, ANN was computed with the ANN toolbox in MATLAB software.

### 3.2. ANFIS Modeling

Constructing the network structure or Takagi-Sugeno fuzzy inference system has been conducted by “if-then” rules of the ANFIS model [56]. As presented in Figure 2, the design of the ANFIS model comprises five layers. The input membership functions (M.F.s) in the first layer transport the inputs to a fuzzy set. Then, in the second layer, fixed nodes are used to measure the firing strength. The firing strength is a quantity to which a fuzzy rule’s antecedent part is fulfilled and determined by an AND or operation, and it shapes the output function for the rule. After that, in the third layer, firing strength values are completed by the normalization process. The parameter set is multiplied by the output of the prior layer to determine the impact of the output. The summation of each input’s signal is determined in the fifth layer [57].

As shown in Figure 2, Pretreatment-1 (acidic or alkaline cleaning: pH (1)), Pretreatment-2 (acidic or alkaline cleaning: pH (2)), deposition time in the cerium bath, and charge transfer resistance of coatings were considered as the inputs and the output respectively for the ANFIS model. According to Table 1, when Step 1 or Step 2 was not applied, the pH value for the step was considered as neutral (pH = 7).

Grid partition (G.P.) is one of the methods for producing the ANFIS structure from available data. It has been reported that as the number of inputs increase, the number of fuzzy rules grows exponentially. In addition, a G.P. technique is only appropriate for a dataset of less than six input parameters [57,58]. In this study, we selected ANFIS-GP, in which gauss M.F. for each input parameter and linear type of M.F. for output with 12 rules were used to construct the fuzzy model.

### 3.3. SVMR Technique

Regression analysis is a strategy for modeling the relationship between dependent and independent variables. Moreover, regression usually finds the linear relationship of variables which may be unique for data analysis. Several polynomial regression models, such as the response vector, parameter vector, and design matrix, create a random error vector. The nonlinear relationship of the variables can be found by polynomial regression [59]. On the other hand, support vector machine regression (SVMR) is a statistical learning theory used for regression and classification of engineering problems. The principles and theoretical background of SVMR can be found in the literature [60,61]. Therefore, for modeling the experimental data in the present research work, Pretreatment-1 (acidic or alkaline cleaning: pH (1)), Pretreatment-2 (acidic or alkaline cleaning: pH (2)), deposition time in the cerium bath, and charge transfer resistance of coatings were considered as the inputs and the output respectively for the SVMR model. According to Table 1, when Step 1 or Step 2 was not applied, the pH value for the step was considered as neutral (pH = 7). In order to find the suitable models and predict the corrosion behavior of CeCCs, a SVMR model will be compared with ANN and ANFIS models.

## 4. Results and Discussion

### 4.1. Different Pretreatments on AA5083 before Applying CeCC

In this section, the effect of different pretreatments on the formation of cerium conversion coating applied to AA5083 is investigated, and corrosion behavior of coated AA5053 plates is studied in 3.5 wt.% NaCl. The goal is to find how different pretreatments influence the formation of the cerium conversion layer and how it controls the corrosion of AA5083. Figure 3 shows the electrochemical polarization curves of cerium coatings applied on the AA5083 samples prepared by different pretreatment methods.

Some features are visible from the obtained curves. Compared with the bare AA5083, all the other coated plates shifted their potential towards more negative values. Compared with the bare AA5083, the other coated plates shifted the cathodic branches toward lower current densities [34,62].

Among the coated samples, the maximum shift of the cathodic branch is related to the coated AA5083 which was pretreated by the D pretreatment method. Even in the cathodic branch of AA5083, oxygen limitation is visible, but this is not revealed in the coated panels. This means CeCC affected the cathodic branches of the coated AA5083. These facts mean that the cerium affected the cathodic branch and suppressed the cathodic reaction, oxygen reduction in neutral NaCl solutions. As mentioned previously, cathodic sites of AA5083 are intermetallic components around the particles, and severe local corrosion has happened. It can be reported that the cerium coating forms a protective layer against the aggressive environment, especially on cathodic intermetallics. It is in agreement with the other findings of the authors and puts in evidence the nature of the corrosion inhibition afforded by the cerium conversion layer [63]. The corresponding corrosion parameters obtained from the polarization curves are listed in Table 2. The data are obtained from the intersection of anodic and cathodic Tafel lines.

From the Table, looking at the Ecorr values, compared with the bare AA5083, all the cerium coated samples presented more negative corrosion potential indicting cerium coating precipitates in cathodic sites such as some intermetallic particles as cathodic current density has been more affected and suppressed [35].

Furthermore, the icorr values revealed that corrosion current density of all the coated samples decreased. As the corrosion current density is considered to be a corrosion rate criterion, the results showed applying CeCC coating on AA5083 pretreated by the D regime led to lower corrosion of the AA5083 substrate. In other words, the D regime could be more effective for formation of CeCC and could be useful in corrosion protection of AA5083. On the other hand, comparing the Ecorr and icorr data showed that cerium coatings applied on the pretreated AA5083 could be more effective than the sample with no pretreatment (A Pretreatment Method). The lowest corrosion current density is obtained for the sample prepared by alkaline pretreatment then acid washing (E Pretreatment Method) before applying the cerium coating. Compared to the bare AA5083, icorr of this pretreated sample was decreased by ~3 orders of magnitude, and in comparison, to the sample with no pretreatment, it has been reduced by more than 1 order of magnitude. It could be indicative that pretreatment could be more effective.

Looking at the anodic and cathodic Tafel slopes, all the coated plates showed different cathodic slopes compared with bare AA5083. This is probably related to the fact that cerium acts in cathodic sites [35,62]. A little change in anodic slopes is related to the change in the anodic reaction of the 5083 Al alloy where the Ce deposited as an oxide/hydroxide and to an interaction between the Ce cation and the Al_2_O_3_ layer present on the alloy surface. As reported in the literature, deposition results from local pH changes produced by the cathodic reaction. During the process, the Al_2_O_3_ layer is destabilized due to the peroxide that changes the pH of the solution. In these conditions, Al^3+^ is hydrolyzed to form different hydroxide complexes [64].

Moreover, local pH increases at cathodic sites on the surface promote the reaction of the Ce (IV) peroxo species with OH^−^ to form a Ce (IV) hydroxyl peroxo species, which with time could decompose to CeO_2_. The cathodic reactions that result in coating deposition are balanced by anodic reactions on the metal surface where aluminum ions are created, suggesting that their presence may influence the coating composition and the deposition process. Therefore, complex cerium (III)/(IV) aluminum oxide/hydroxide layer may be deposited on the aluminum surface during the conversion treatment of the alloy.

EIS was performed to study the effect of different pretreatment conditions on the cerium coatings applied on AA5083. Typical Nyquist and Bode plots of cerium coated AA5083 alloy obtained in the other pretreatment conditions are shown in Figure 4.

According to the Nyquist plots, it is clear that coated plates showed larger semicircles than the bare ones. Moreover, the cerium coated sample without any pretreatment showed much lower corrosion resistance than the pretreated samples. It indicates that a pretreatment before coating deposition could be more effective as it prepares a more active surface for deposition. Among the pretreated samples, the sample prepared by the pretreatment of alkaline then acid washing (D Pretreatment Method) presented a larger semicircle in Nyquist, a higher module in Bode, and a broader peak in the phase diagram that means higher corrosion protection given by the sample. This fact is probably due to the more effective surface coverage of this sample than the others. For a deep study of corrosion protection, experimental impedance spectra were fitted using the different equivalent circuits shown in Figure 5. In the figures, Rs is NaCl solution resistance as corrosive media; in the case of coated plates, Rf is cerium coating resistance, while in the case of AA5083, it is related to the natural oxide film. Rct is the charge transfer resistance of the AA5083 indicating corrosion rate [65]. Due to the non-ideal behavior of the capacitors, constant phase elements (CPE) were used instead of capacitors. In capacitors, the phase angle is −90, while in this phase diagram, angles are lower than −90. Therefore, CPEf and CPEdl are constant phase elements of cerium film and an electrical double layer formed on the AA5083 surface, respectively. This parameter consists of n and Y0, the power of impedance and admittance (reverse of impedance), respectively.

In the case of AA5083 and the coated AA5083 prepared by no pretreatment, the equivalent circuit consisted of an inductor; this is due to the presence of a passive layer on the surface. Ws considered for the C and E pretreatment methods which are related to mass transport limitation by diffusion. The EIS parameters were obtained from fitting of EIS diagrams of cerium coated AA5083 samples in 3.5 wt.% NaCl corrosive solution provided by different pretreatment conditions are reported in Table 3.

Looking at the cerium coating resistance values, the highest Rf is related to the sample of the D Pretreatment Method. Moreover, this sample also presents the highest charge transfer resistance (Rct) at the charge transfer resistance values. All of the pretreated samples showed the Rct values much more than AA5083. No significant difference was observed between the AA5083 and the one coated with no pretreatment (A Pretreatment Method). This information shows that there is a good agreement between D.C. polarization and EIS data. The following explanations can be implemented. As mentioned previously, the main reason for corrosion of AA5083 in NaCl solutions is related to the presence of different intermetallics distributed in the matrix, leading to local corrosion and pitting. The known distributed intermetallics are Al (Mn, Fe, Cr), Al6(Fe, Mn), or iron-rich intermetallics. Most corrosion occurs around these cathodic intermetallics causing the formation of a pit with high current density [66].

Other intermetallics forming in AA5083 are Al (Si, Mg) and Al-Mg, which are more active than the matrix and having an anodic nature. No pit is formed around these particles, but magnesium’s high reactivity leads to the dissolution and the fast dealloying of these intermetallics. However, hydroxide (Mg(OH)_2_ and SiO_2_.nH_2_O) precipitations formed during corrosion act as a diffusion barrier suppressing the deeper pits [9].

Another type of corrosion of AA5083 in NaCl is related to the destructive effect of Cl^−^ that destroys the oxide film of the matrix. However, corrosion around the metal particles with cathodic nature is more concerning and should be considered [66,67].

With these descriptions, since cerium acts as a cathodic inhibitor, it can effectively reduce the influence of these cathodic intermetallics [35,61].

Cerium reacts with OH^−^ generated by oxygen reduction on the cathodic area such as the intermetallic surface. Therefore, it can significantly reduce alkaline attacks around the particles. Therefore, in the case of the cerium coated sample with no pretreatment, since the rest of the surface is probably covered with an oxide nanometer layer, the cathodic intermetallic surfaces are the only suitable place for the deposition of the cerium layer [35,49,61].

Of course, it is worth noting that destructive chlorine ions attack the nanometer oxide layer, and these two events determine the corrosion resistance of this specimen. The deposition of the cerium layer on the cathodic intermetallics further increases the corrosion resistance of this sample compared to AA5083. The coating resistance of this sample is the sum of the coating resistance of the oxide layer on the matrix and this cerium layer on the cathodic intermetallics surface. However, it is different for the other coated samples, and the effect of acid washing and alkaline cleaning on the surface preparation should be investigated. Due to the high tendency of Al to oxidize, the surface of aluminum is mainly covered by a nanometer oxide layer. It is almost impossible to form a cerium layer on this oxide layer and it should be removed [9,66].

This oxide is dissolved in pH above ~11.5 and forms aluminate (AlO^2−^) ions. Hence, alkaline etching changes the nature of the oxide layer, and then the AlO^2−^ ions react with cation cerium ions to form a conversion layer [9]. Below pH ~4, in the acidic pH range, Al^3+^ is stable, and the oxide layer is dissolved. In the range of 4–11.5, Al_2_O_3_ is stable. It seems that acid washing can provide a matrix more suitable for cerium oxide/hydroxide precipitation. Comparing Rf and Rct confirmed that corrosion resistance of acid washing and alkaline then acid washing (D Pretreatment Method) provided a more active surface for deposition of the cerium layer and provides a more appropriate substrate for film formation and makes a film more uniform. The effect of alkaline or acid washing is on the matrix and its power to remove or alter the oxide formed on the aluminum substrate.

The SEM images of surface morphology of the uncoated AA5083 and coated samples of A, B, C, D, and E pretreatment methods are shown in Figure 6. The energy dispersive spectroscopy (EDS) of the coated sample prepared under different conditions is shown in Figure 7 and Table 4. The best surface coverage was presented by the D Pretreatment Method (Figure 6e). Cracks are observed by some authors studying the cerium layer on aluminum [34,67,68,69,70].

When the surface is only degreased by acetone (A Pretreatment Method), it can be concluded that its effects are just degreasing on the surface, and this cannot provide a proper surface for precipitating the cerium and cerium deposits, probably only on the cathodic intermetallic. Therefore, local precipitation was observed. SEM images agreed with the EIS data as the best corrosion protection performance was presented by the D pretreatment method and the lowest by the no pretreatment sample (Method A). The reason that the resistance of the coatings in the impedance test was low is due to the presence of cracks and these local deposits. In the case of the coated samples, we attempted to take an element analysis from the deposits or the coating area. For this reason, some of the elements in the base metal are not seen in their spectra.

### 4.2. Effect of Different Deposition Times for D Pretreatment Method

According to the previous section, the sample which was pretreated in the alkaline solution then washed in acid (D Pretreatment Method) showed a more uniform film with higher protection performance. The anti-corrosion performance of the cerium coating deposited polarization curve was carried out for coated AA5083 specimens to find the effect of deposition time. The D Pretreated Method was performed with a deposition time of 1-, 5-, 10-, and 20-min. Figure 8 shows the electrochemical polarization curves of coated AA5083 in 3.5 wt.% NaCl solution at different deposition times in the cerium bath. All the pretreated samples shifted their curves to more negative values. It means that the cerium coating could be considered a cathodic inhibitor, causing a decrease in the cathodic potential. On the other hand, all cathodic branches were shifted to the left, where lower cathodic current density had been revealed. 

The corresponding corrosion parameters obtained from polarization curves are listed in Table 5. The acquired data revealed that the corrosion current densities (icorr) decrease with the increase of deposition time in the coating solution up to 10 min; then, the value declined due to the higher thickness of the formed coating, leading to more cracks. In other words, the formation and propagation of microcracks are related to the film thickness’s growth, leading to decreased anti-corrosion performance and increased corrosion rate of this sample.

The EIS technique was employed to study the influence of deposition time for coated and uncoated AA5083 samples prepared by the D Pretreatment Method with different deposition times (1, 5, 10, and 20 min). The curves were recorded after about 1 h of immersion in 3.5 wt.% NaCl corrosive electrolyte. Different models of impedance spectroscopy are plotted in Figure 9a–c.

Comparison of Nyquist plots shows that the semicircle diameter increases as the deposition time increases during the initial 10 min and then decreases. In addition, looking at the Bode and phase diagram, the highest Bode-module and broader relaxation time is related to the coated AA5083 pretreated for 10 min in the D Pretreatment Method. For the times up to 10 min, the thickness of the cerium film increases, but for longer deposition times, due to the formation of a thicker film of cerium and the creation of more cracks within the film, the protection performance of the coating reduces.

The equivalent circuit shown in Figure 5b was used for modeling the electrical behavior of these samples. This observation is well compatible with the Tafel results and again confirms that longer deposition times lead to a more uniform coating formation, but up to a critical value. The gained fit data are reported in Table 6. In shorter deposition times, the film was not continuous enough to cover the whole surface of the substrate. The best deposition time was obtained for 10 min of deposition. After this critical time, a cracked film formed on the surface that could not be protective.

### 4.3. Modeling Results

As mentioned in the modeling details, the purpose is to provide an optimized model for simulating the Rct of CeCCs. As demonstrated in Figure 10, increasing the neurons in the hidden layer from 2 to 10 resulted in a variation of the ANN model performance. Variation of *R*^2^ was observed by changing the neurons in the hidden layer from 2 to 10. Four-neuron hidden layer architecture was used as it led to the highest *R*^2^ values resulting in the most trustable results (*R*^2^ = 0.9). Accordingly, for further optimization of the training phase, a single hidden layer that consists of four neurons and the transfer functions were used. For this, six different ANN structures were developed to obtain a suitable training algorithm to simulate the corrosion behavior of CeCCs. The models’ MSE and MAE values were compared to construct the optimal ANN model (Table 7). The model that used the L.M. training algorithm was chosen since it has the lowest MSE, MAE (1048.3, 14.1), and a higher coefficient of determination than the other models (0.90).

Figure 11 shows the relation between experimental (observed) and modeled (predicted) amounts of corrosion resistance of CeCCs achieved by ANN, ANFIS, and SVMR models. Generally, the accumulation of data points around the Y = X line indicates that the anticipation of the experimental data is achieved with higher accuracy. The points near the regression line proved the efficiency of the developed model. In addition, the amount of *R*^2^ for ANN, ANFIS, and SVMR models were 0.99, 90, and 0.69 respectively, and proved that all the models could simulate the Rct of CeCCs. The values of MSE and MAE for the ANFIS model were 48.83 and 3.49 respectively. Based on Table 8, even though ANN and SVMR models displayed a reliable performance, it is explicit that the predictive potential of the ANFIS model is higher than the other models.

Several parameters were utilized to evaluate the ability of each model to validate the constructed computational models. Smith et al. [70] reported the R formula to assess the performance of a built model. The equations are provided in Table 9. In this equation, hi and ti represents the observed and predicted outputs, respectively, and the average value for corresponding parameters. Moreover, other statistical criteria were calculated for the models to compare the performance. Where k is the slope of the regression line which is obtained from actual output (hi) against anticipated output (ti) and k’ is the anticipated output versus actual output (ti Vs. hi) [30,71]. As presented in this Table, other functional parameters, such as m, n, and Rm, were calculated for the models [72,73,74]. The results demonstrated that computational models could satisfy the standard condition for each criterion. However, the performance of the ANFIS model was significantly better compared to the other models. Therefore, the ANFIS is the most accurate model for simulating the corrosion behavior of CeCCs.

The ANFIS model is used for studying the effect of input parameters on the Rct of CeCCs (Figure 12). In order to observe the effect of each parameter, other parameters were assumed to have a constant value. The results demonstrated that the Rct was significantly affected by these parameters. Generally, an increase in pH (1) (Figure 12a) results in the higher Rct of CeCCs. In contrast, increasing the pH (2) decreases the Rct. The optimum value for deposition time was 10 min. These achievements are in agreement with the experimental results.

Finally, the ANFIS model was used in order to simulate the effect of inputs parameters on the corrosion behavior of CeCCs. As illustrated in Figure 13, the corrosion behavior was greatly influenced by input variables. These modeling results agree with the experiments’ outcomes, in which the best corrosion resistance of CeCCs was achieved at pH (1) = 12, pH (2) = 3, and a deposition time of 10 min.

## 5. Conclusions

In this research, cerium-based conversion coatings were applied on an AA5083 aluminum alloy, and the effect of surface pretreatments on the corrosion resistance and morphology of the coatings was studied. The pretreatment of the alkaline solution then acid washing presented higher corrosion protection compared to other pretreatment methods. The outcome of the polarization test demonstrated that the corrosion current density of alkaline solution then acid washing pretreatment is significantly reduced compared to the sample without pretreatment. EIS results confirmed that the corrosion resistance of alkaline then acid washing (D Pretreatment Method) provided a more active surface for the deposition of the cerium layer and provided a more appropriate substrate for film formation, and made a film more uniform. Moreover, the best surface coverage and higher deposition of cerium layer on the aluminum surface was observed with the alkaline solution then acid washing pretreatment method. After selecting the best surface pretreatment, various deposition times of cerium baths were investigated. The results demonstrated that the best deposition time was obtained for 10 min of deposition, and after this critical time, a cracked film formed on the surface that could not be protective. ANN, ANFIS, and SVMR computational models were used to simulate the Rct of cerium-based conversion coatings as a function of Pretreatment-1 (acidic or alkaline cleaning: pH (1)), Pretreatment-2 (acidic or alkaline cleaning: pH (2)), and deposition time in the cerium bath. Different ANN models were constructed by changing the parameters, and the most suitable ones were selected. Finally, an ANN model with a MAE value of 14.10 was achieved, with four neurons and Levenberg–Marquardt backpropagation as a training algorithm. The mean absolute error (MAE) of the fuzzy model was 3.49. The subsequent validity and performance analysis of the models with several statistical criteria such as *R*, k, k′, *m*, n, *Rm*, (*R*0)^2^ and (*R*0′)^2^ verified that the ANFIS model was more robust than the other models. *R*^2^ of the ANN, ANFIS and SVMR models was 0.90, 0.99, and 0.69, respectively. Using the ANFIS model, the Rct of cerium-based conversion coatings was simulated with higher accuracy and precision.

## Figures and Tables

**Figure 1 molecules-26-07413-f001:**
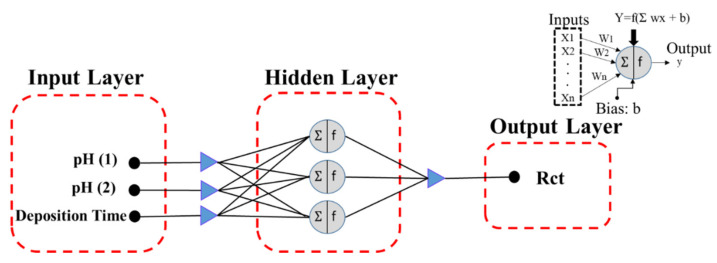
The structure for the ANN model utilized for modeling the corrosion behavior of CeCCs.

**Figure 2 molecules-26-07413-f002:**
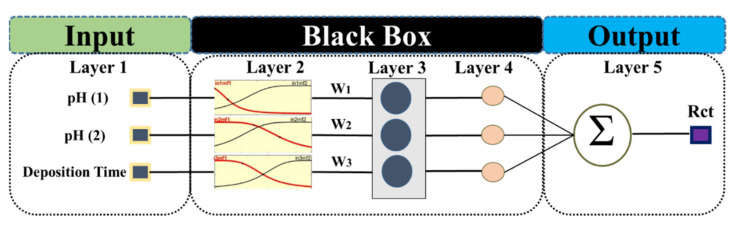
Typical structure of the ANFIS model for modeling the corrosion behavior of CeCCs; the model comprises 3 inputs and 5 layers.

**Figure 3 molecules-26-07413-f003:**
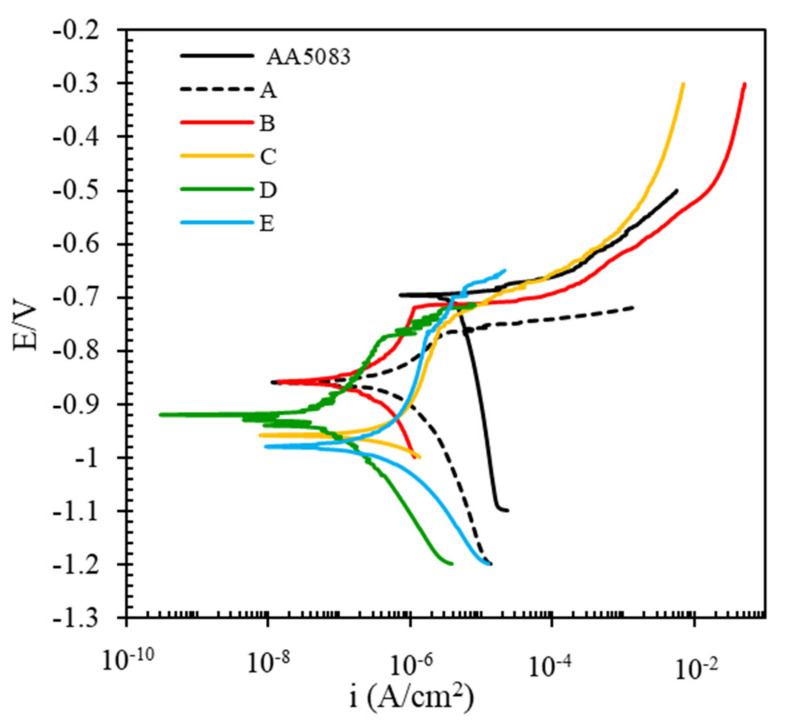
The electrochemical polarization curves of cerium coated AA5083 were prepared under different pretreatment conditions (corrosive solution 3.5 wt.% NaCl).

**Figure 4 molecules-26-07413-f004:**
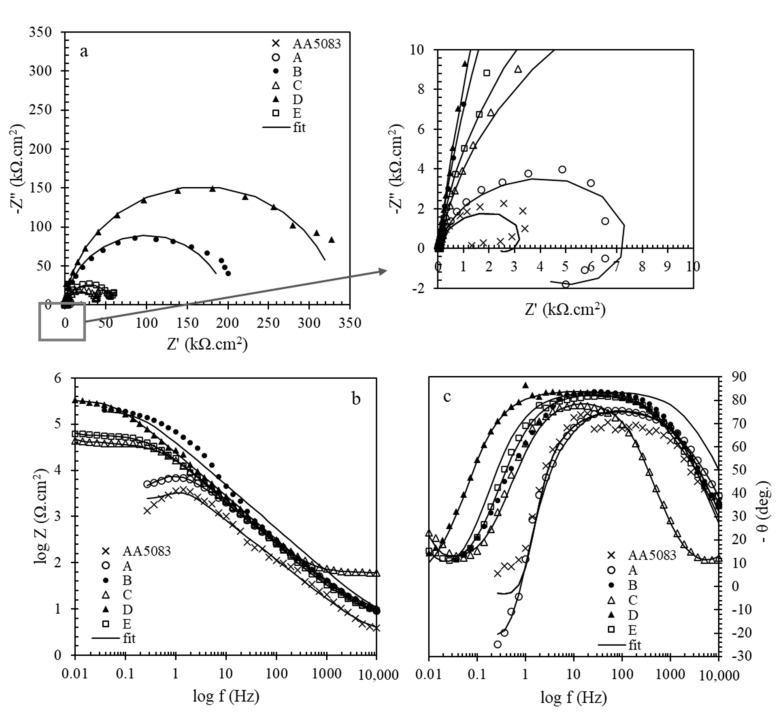
Typical (**a**) Nyquist, (**b**) Bode and (**c**) phase diagrams of cerium coated Al 5083 samples in 3.5 wt.% NaCl corrosive solution provided by different pretreatment conditions.

**Figure 5 molecules-26-07413-f005:**
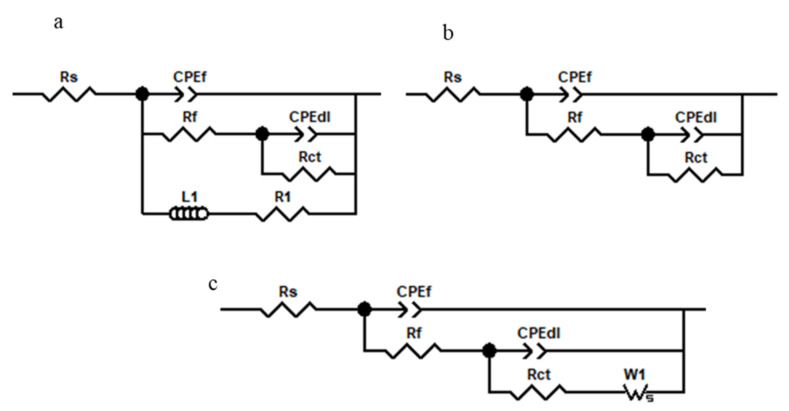
Different equivalent circuits used to fit EIS diagrams of cerium coated AA5083 samples in 3.5 wt.% NaCl corrosive solution provided by different pretreatment conditions. (**a**) Used for AA5083 and without pretreatment aluminum plate (A Method), (**b**) used for the coated plates pretreated with acid pickling (B Method) and alkaline etching followed by acid washing (D Method), and (**c**) used for the coated plates pretreated with alkaline etching (C Method) and acid washing followed by alkaline etching (E Method).

**Figure 6 molecules-26-07413-f006:**
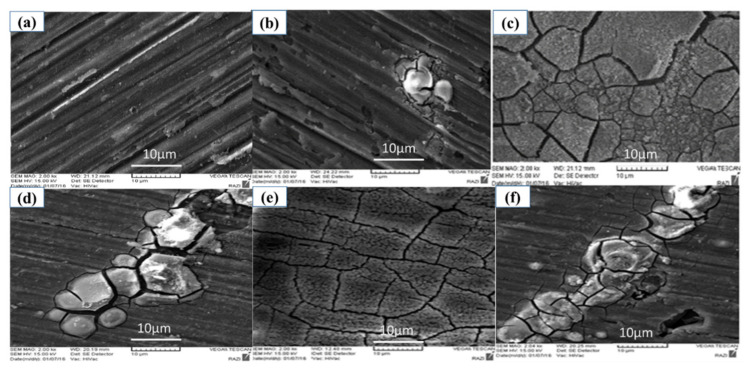
SEM micrographs of different cerium coated and uncoated AA5083 prepared by various pretreatment methods (**a**) uncoated AA5083, (**b**) A, (**c**) B, (**d**) C, (**e**) D, and (**f**) E.

**Figure 7 molecules-26-07413-f007:**
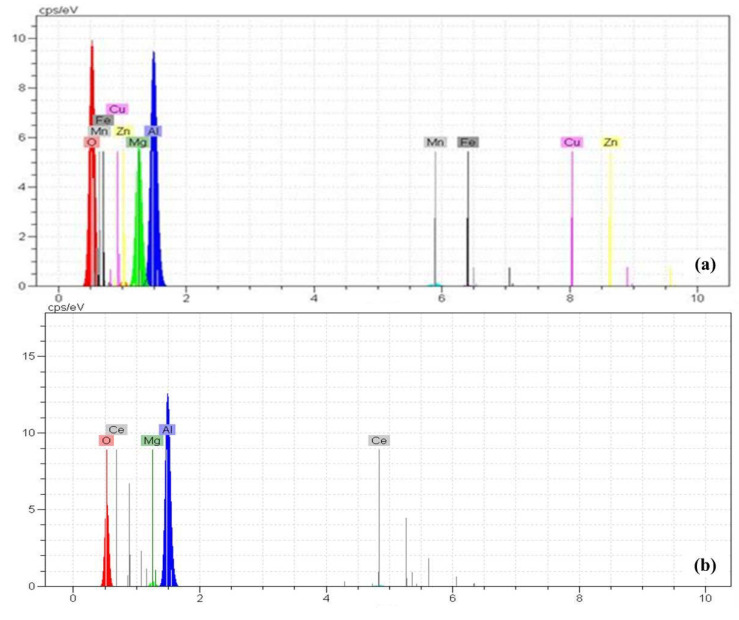
EDS analysis of different cerium coated and uncoated AA5083 prepared by different pretreatments (**a**) uncoated AA5083, (**b**) A, (**c**) B, (**d**) C, (**e**) D, and (**f**) E.

**Figure 8 molecules-26-07413-f008:**
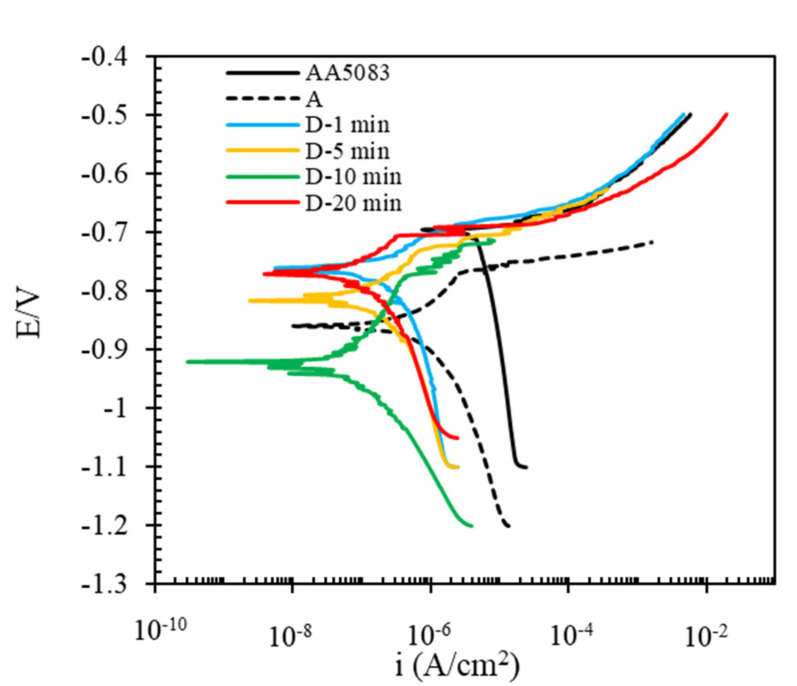
Polarization curves of coated and uncoated AA5083 samples in 3.5 wt.% NaCl corrosive media. The samples were prepared by pretreatments of washing at different deposition times (1, 5, 10, and 20 min) for the D Pretreatment Method.

**Figure 9 molecules-26-07413-f009:**
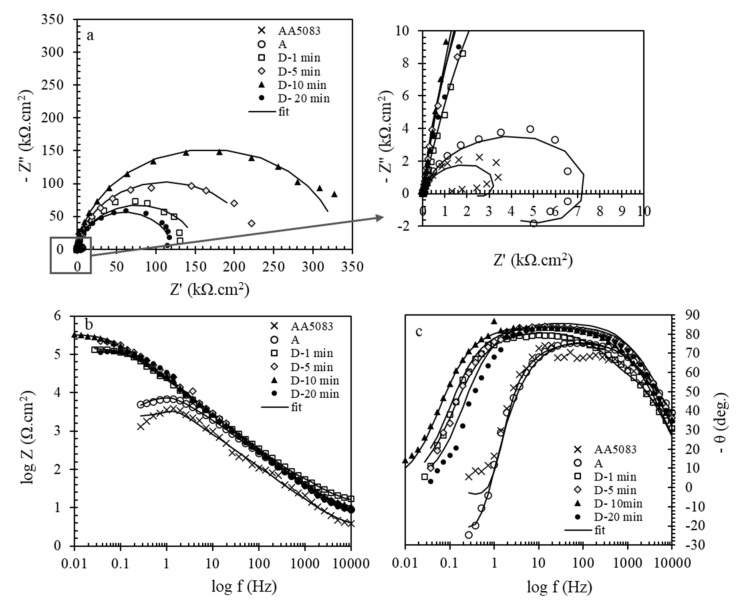
Typical (**a**) Nyquist, (**b**) Bode and (**c**) phase plots of coated and uncoated AA5083 samples in 3.5 wt.% NaCl corrosive media. The samples were prepared by the D Pretreatment Method with different deposition times (1, 5, 10, and 20 min).

**Figure 10 molecules-26-07413-f010:**
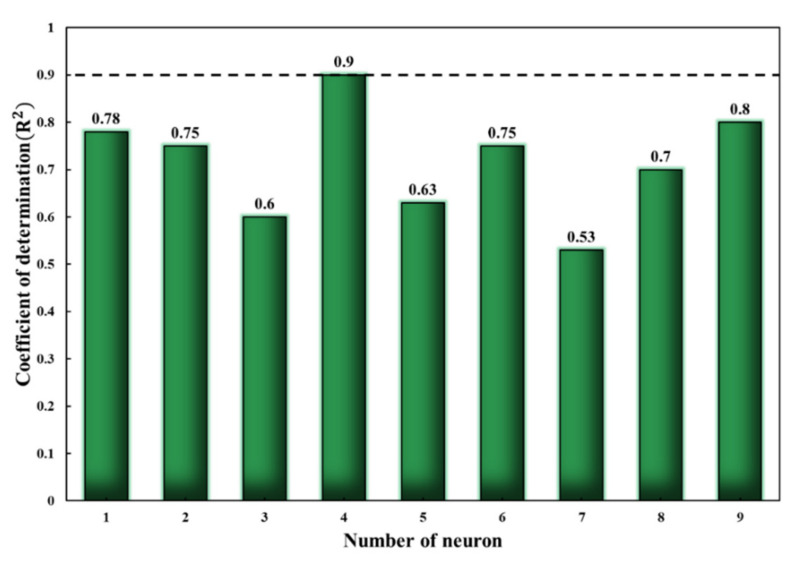
The variation of *R*^2^ for the ANN model with the different number of neurons from 2 to 10 in the hidden layer.

**Figure 11 molecules-26-07413-f011:**
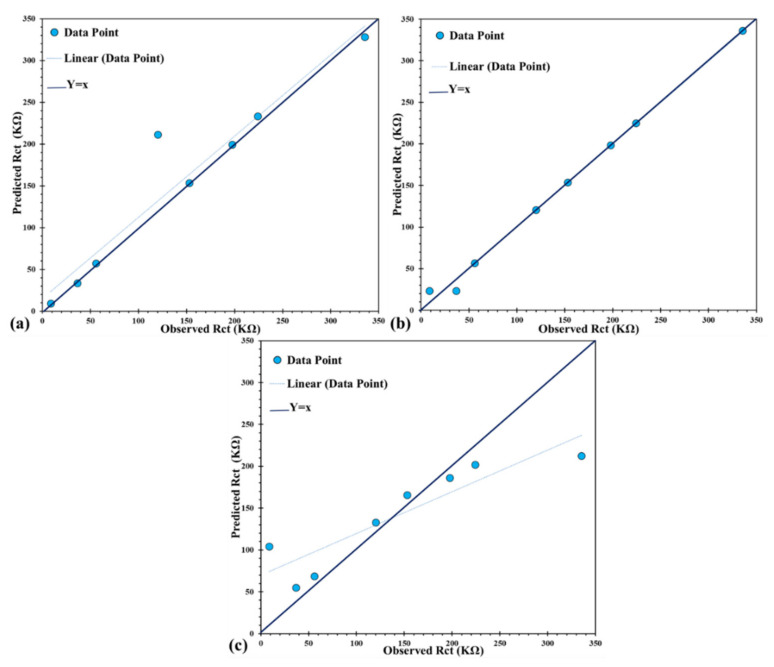
The correlation of the observed and predicted Rct of CeCCs obtained by the computational models: (**a**) ANN, (**b**) ANFIS, and (**c**) SVMR. The dashed line shows a fitted line to the modeling result, and solid lines represent the line with a slope of one.

**Figure 12 molecules-26-07413-f012:**
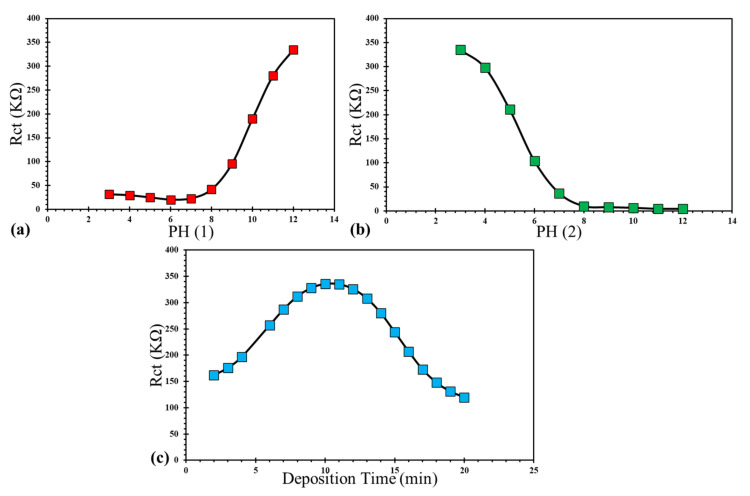
The effect of (**a**) pH (1), (**b**) pH (2), and (**c**) deposition time on the Rct of CeCCs obtained from the ANFIS model. Each diagram is plotted by assuming a constant value for other parameters. The assumed constant value of the pH (1), pH (2), deposition time: 12, 3, and 10 min.

**Figure 13 molecules-26-07413-f013:**
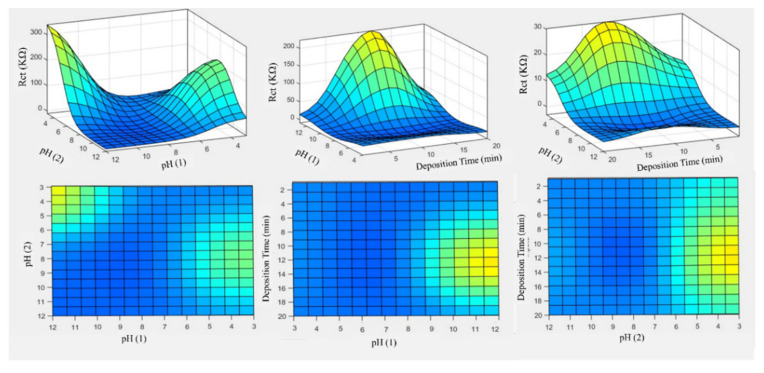
The 3D and 2D plots were obtained from the ANFIS model for Rct of CeCCs.

**Table 1 molecules-26-07413-t001:** Different surface pretreatment applied on AA5083 alloy.

Sample Code	Step 1	Step 2	Duration of Pretreatment (s)
A	No	No	-
B	Acidic solution (1N H_2_SO_4_)	No	30
C	Alkaline solution (1N NaOH)	No	30
D	Alkaline solution (1N NaOH)	Acidic solution (1N H_2_SO_4_)	30
E	Acidic solution (1N H_2_SO_4_)	Alkaline solution (1N NaOH)	30

**Table 2 molecules-26-07413-t002:** The extracted parameters from the polarization curves of different pretreated AA5083 coated with CeCC.

Sample	Ecorr (V)	Icorr (A/cm^2^)	ba (V/dec)	-bc (V/dec)
bare AA5083	−0.639	3.088 × 10^−5^	0.047	2.96
A	−0.860	7.911 × 10^−7^	0.265	0.123
B	−0.906	9.823 × 10^−8^	0.172	0.112
C	−0.969	3.048 × 10^−7^	0.036	0.167
D	−0.927	4.085 × 10^−8^	0.078	0.148
E	−0.979	2.166 × 10^−7^	0.153	0.067

**Table 3 molecules-26-07413-t003:** EIS parameters obtained from the fitting of EIS diagrams of the coated Al 5083 samples in 3.5 wt.% NaCl corrosive solution provided by different pretreatment methods. Bare AA5083 is considered as a reference.

Code of Samples	Rs (Ω·cm^2^)	Rf (Ω·cm^2^)	CPEf		Rct (Ω·cm^2^)	CPEdl		L1 (H·cm^2^)	R1 (Ω·cm^2^)	W1-R	W1-T	W1-P
			n	Y0 (Ω^−1^·cm^−2^ sn)		n	Y0 (Ω^−1^·cm^−2^ sn)					
AA5083	2.9	44.0	0.86	3.32 × 10^−5^	5467.0	0.98	2.26 × 10^−7^	858.5	4271.0	-	-	-
A	5.5	58.7	0.93	3.96 × 10^−6^	8958.0	0.84	4.02 × 10^−6^	3517.0	5254.0	-	-	-
B	5.5	106.1	0.93	3.96 × 10^−6^	198,000.0	0.97	3.94 × 10^−7^	-	-	-	-	-
C	1.0	62.4	0.92	1.07 × 10^−7^	36,909.0	0.92	9.45 × 10^−6^	-	-	34,305.0	55.46	0.65
D	8.6	300.2	0.93	6.67 × 10^−6^	335,730.0	0.99	7.32 × 10^−7^	-	-	-	-	-
E	6.9	100.9	0.92	8.54 × 10^−6^	56,190.0	0.97	156 × 10^−6^	-	-	22,098.0	42.42	0.67

**Table 4 molecules-26-07413-t004:** Percentage of the element of EDS analysis of different cerium coated and uncoated AA5083 prepared by different pretreatments.

Code of Sample	Al	Mg	Ce	Mn	Fe	Cu	Zn	O
AA5083	93.4	4.3	-	0.8	0.4	0.1	0.2	0.3
A	67.58	1.05	1.05	-	-	-	-	30.32
B	30.74	1.20	24.45	-	-	-	-	43.61
C	67.99	1.68	7.58	-	-	-	-	21.74
D	14.74	-	30.58	-	-	-	-	50.07
E	68.21	1.43	9.91	-	-	-	-	20.44

**Table 5 molecules-26-07413-t005:** Data gained from the polarization curves of coated and uncoated AA5083 samples in 3.5 wt.% NaCl corrosive media. The samples were prepared by the D Pretreatment Method with the different deposition times of (1, 5, 10, and 20 min).

Sample	Ecorr (V)	Icorr (A/cm^2^)	ba (V/dec)	-bc (V/dec)
AA5083	−0.639	3.088 × 10^−5^	0.047	2.96
A	−0.860	7.911 × 10^−7^	0.265	0.123
D-1 min	−0.762	9.552 × 10^−8^	0.122	0.031
D-5 min	−0.817	5.624 × 10^−8^	0.090	0.050
D-10 min	−0.927	4.085 × 10^−8^	0.078	0.148
D-20 min	−0.749	9.873 × 10^−8^	0.122	0.023

**Table 6 molecules-26-07413-t006:** Obtained fit data of EIS diagram of assessing the effect of deposition time in D Pretreatment Method.

Code of Samples	Rs (Ω·cm^2^)	Rf (Ω·cm^2^)	CPEf		Rct (Ω·cm^2^)	CPEdl		L1 (H·cm^2^)	R1 (Ω·cm^2^)
			n	Y0 (Ω^−1^·cm^−2^ sn)		n	Y0 (Ω^−1^·cm^−2^ sn)		
AA 5083	2.9	44.0	0.86	3.32 × 10^−5^	5467.0	0.98	2.26 × 10^−7^	858.5	4271.0
A	5.5	58.7	0.93	3.96 × 10^−6^	8958.0	0.84	4.02 × 10^−6^	3517.0	5254.0
D-1 min	13.2	154.9	0.92	5.17 × 10^−6^	153,200.0	0.91	3.11 × 10^−6^	-	-
D-5 min	6.3	200.0	0.94	6.38 × 10^−6^	224,520.0	0.97	4.05 × 10^−7^	-	-
D-10 min	8.6	300.2	0.93	6.67 × 10^−6^	335,730.0	0.99	7.32 × 10^−7^	-	-
D-20 min	6.3	100.0	0.96	5.80 ×1 0^−6^	120,390.0	0.96	7.27 × 10^−6^	-	-

**Table 7 molecules-26-07413-t007:** The effect of the training algorithm on the performance of the six ANN models executed with four neurons in the hidden layer architecture.

ANN Models	Training Algorithm	Symbol	MAE	MSE
ANN-1	Resilient backpropagation	RP	38.5	2604.6
ANN-2	BFGS quasi-Newton backpropagation	BFG	58.4	4884.1
ANN-3	Scaled conjugate gradient	SCG	37.1	2676.9
*ANN-4*	*Levenberg–Marquardt backpropagation*	*LM*	*14.1*	*1048.3*
ANN-5	Gradient descent with momentum and adaptive LR	GDX	51.7	4267.5
ANN-6	Conjugate gradient with Powell/Beale restarts	CGB	37.1	2676.9

The best model is shown in italics.

**Table 8 molecules-26-07413-t008:** The performance results for constructed models.

Model	ANN	ANFIS	SVMR
Error			
MSE	1048.27	48.83	3220.58
MAE	14.10	3.49	38.55
*R* ^2^	0.90	0.99	0.69

**Table 9 molecules-26-07413-t009:** The validation and performance of constructed models.

**Item**	**Formula**	Condition	ANN	ANFIS	SVMR	Item	Formula
1	*R* $\frac{\sum_{i=1}^{n}\left(hi-\bar{hi}\right)\left(ti-\bar{ti}\right)}{\sqrt{\sum_{i=1}^{n}{\left(hi-\bar{hi}\right)}^{2}}\sqrt{\sum_{i=1}^{n}{\left(ti-\bar{ti}\right)}^{2}}}$	0.8 < *R*	0.957	0.998	0.857	1	*R* $\frac{\sum_{i=1}^{n}\left(hi-\bar{hi}\right)\left(ti-\bar{ti}\right)}{\sqrt{\sum_{i=1}^{n}{\left(hi-\bar{hi}\right)}^{2}}\sqrt{\sum_{i=1}^{n}{\left(ti-\bar{ti}\right)}^{2}}}$
2	k = $\frac{\sum_{i=1}^{n}\left(hi\times ti\right)}{\sum_{i=1}^{n}hi^{2}}$	0.85 < k < 1.15	1.042	0.998	0.862	2	k =$\;\frac{\sum_{i=1}^{n}\left(hi\times ti\right)}{\sum_{i=1}^{n}hi^{2}}$
3	k′ = $\frac{\sum_{i=1}^{n}\left(hi\times ti\right)}{\sum_{i=1}^{n}ti^{2}}$	0.85 < k′ < 1.15	0.932	1.001	1.097	3	k′ = $\frac{\sum_{i=1}^{n}\left(hi\times ti\right)}{\sum_{i=1}^{n}ti^{2}}$
4	m =$\;\frac{R^{2}-R0^{2}}{R^{2}}$	*m* < 0.1	0.091	0.0047	0.098	4	m = $\frac{R^{2}-R0^{2}}{R^{2}}$
5	*n* = $\frac{R^{2}-{Ro}^{\prime 2}}{R^{2}}$	*n* < 0.1	0.096	0.0046	0.099	5	N = $\frac{R^{2}-Ro{’}^{2}}{R^{2}}$
6	*Rm* =$\;R^{2}\;$× (1 − $\sqrt{\left|R^{2}-R0^{2}\right|}$)	0.5 < *Rm*	0.623	0.928	0.546	6	*Rm* = $R^{2}$× (1 − $\sqrt{\left|R^{2}-R0^{2}\right|}$)
Where	$R0^{2}\;$= 1 − $\frac{\sum_{i=1}^{n}{\left(ti-hi^{0}\right)}^{2}}{\sum_{i=1}^{n}{\left(ti-\bar{\;ti}\right)}^{2}}$, $hi^{0}\;$= k × ti	≈1	0.994	0.999	0.771	Where	$R0^{2}$ = 1 −$\;\frac{\sum_{i=1}^{n}{\left(ti-hi^{0}\right)}^{2}}{\sum_{i=1}^{n}{\left(ti-\bar{ti}\right)}^{2}}$, $hi^{0}$ = k × ti

## Data Availability

Not applicable.

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
