# Peer review of "Surface Pretreatments of AA5083 Aluminum Alloy with Enhanced Corrosion Protection for Cerium-Based Conversion Coatings Application: Combined Experimental and Computational Analysis"

_molecules, 2021, doi:10.3390/molecules26247413_

Round 1
Reviewer 1 Report
Conversion coating is one of the best ways for anticorrosion of metallic materials. In this paper the improvement of conversion coating was achieved by pretreatments. Be comparative study, the best pretreatment was found to be alkaline solution then acid washing, providing a more appropriate substrate for film formation. The best deposition time of 10 min was also obtained. Finally the various parameters affecting the corrosion behavior were studied via computational model. The research is of good reference for combating the corrosion problem. English needs to be modified by a qualified talent.
Author Response
Q1: English needs to be modified by a qualified talent.
Response: I appreciated your comment, The article is given to a native English-speaking person to check for any possible types of grammatical errors. Your comment was applied.
Reviewer 2 Report
The authors of the article compare the effect of surface pretreatments on cerium-based coatings, using experimental data and modeling. The article is very long (has many pages, but it is not review), which makes it difficult to read and comprehend (some data can be combined or shortened or removed). In general, the article is interesting. I have the following comments, which I hope will be helpful for improving the article before it can be recommended for publication in Molecules.
- Surface treatments have been known for a long time and it has been found that alkaline solution then acid pretreatment is the best method for coating. What is the novelty of the work?
- I'm not sure about references 2 and 3 used in lines 55-56 because the authors in references 2-3 describe the β-phase (Mg2Al3), not the Al8Mg5 particles that the authors point out. The structure of the AA5083 alloy consists of a solid solution based on aluminum with a β-phase in the usual case. Check please.
- The authors do not mention in any way how the alloy was obtained and its composition. It is necessary to clarify in what condition and after what heat treatment the samples are used.
- Point 3. Characterization. Characterization of what? It is not clear.
- The choice of material is not discussed. Why do authors use СеСC? It should be clarified.
- Linе 261 «The SVMR model was compared with the other model to achieve an effective technique for predicting the corrosion behavior of CeCCs» - Please clarify what is the other model?
- Line 311: Reference 56 examines the effect of Ce, how can it be compared with CeCC?
- It would be helpful if some explanation about table 2. Icorr, for example, is explained only on line 479. Also the data A5083 and A in table 5 have already been present in table 2. Figure 3 is unclear if compare with figure 8.
- In the chapter Results, the authors cite a lot of references, it is not always clear that the results are taken from the literature or obtained by the authors.
- Surface morphology in Figure 6 requires an explanation. Cracks on the surface raise doubts as to why mode D is better than the others.
- It is necessary to enlarge Fig. 7.
- Why do the data in Fig. 4 and Fig. 9 not match for A?
- It is not clear table 7, what is the fundamental difference in ANN? How much is it necessary?
- Simulation is incomprehensible with what it is compared in Fig. 11
- Please check figure 12.
Author Response
Q1: Surface treatments have been known for a long time and it has been found that alkaline solution then acid pretreatment is the best method for coating. What is the novelty of the work?
Response: effect of alkaline or acidic pretreatment on other alloys of Al had been studied but the effect of the sequence of alkaline and acidic conditions on AA5083 has not been investigated. Intermetallic in these alloys are different from the other types of AA, so we decided to study AA5083. Additionally, the main purpose of this work was to find a good correlation between electrochemical results and ANFIS modeling which no study has been done.
Q2: I’m not sure about references 2 and 3 used in lines 55-56 because the authors in references 2-3 describe the β-phase (Mg2Al3), not the Al8Mg5 particles that the authors point out. The structure of the AA5083 alloy consists of a solid solution based on aluminum with a β-phase in the usual case. Check please.
Response: references 2 and 3 were checked. The statement was changed and some references were added to it.
Q3: 1. The authors do not mention in any way how the alloy was obtained and its composition. It is necessary to clarify in what condition and after what heat treatment the samples are used.
Response: chemical composition of the AA5083 was added to the revised manuscript at the first experimental part. The samples were used without any heat treatment.
Q4: Point 3. Characterization. Characterization of what? It is not clear.
Response: The manuscript was revised. It changed to “surface characterization”.
Q5: The choice of material is not discussed. Why do authors use СеСC? It should be clarified.
Response: Ce3+ ion has been demonstrated to be very effective as a cathodic inhibitor for these alloys. As the main problem in the corrosion of AA5083 is the presence of cathodic intermetallic, the use of cerium has been revealed to be an effective protection system of aluminum alloys. These results have been taken as the starting point for the design of protective systems based on Ce for aluminum alloys. Moreover, some sentences were added to the revised manuscript in lines156-157.
Q6: Linе 261 «The SVMR model was compared with the other model to achieve an effective technique for predicting the corrosion behavior of CeCCs» - Please clarify what is the other model?
Response: In this study, we developed three computational models consisting of Artificial neural network (ANN), adaptive neuro-fuzzy inference system (ANFIS), and support vector machine regression (SVMR). In this statement, other models are ANN and ANFIS. The statement was revised and highlighted in the text.
Q7: Line 311: Reference 56 examines the effect of Ce, how can it be compared with CeCC?
Response: The mechanism of action of the Ce elements is based on blocking the cathodic areas of the material, reducing the rate of the cathodic process and, as a consequence, that of the associated anodic process.
Q8: It would be helpful if some explanation about table 2. Icorr, for example, is explained only on line 479. Also, the data A5083 and A in table 5 have already been present in table 2. Figure 3 is unclear if compare with figure 8.
Response: one paragraph (lines 321-327) has been discussed on icorr of the CeCC coated AA5083 which were made in different pretreatment. However, some sentences were added to the revised manuscript regarding corrosion current density.
- Yes, AA5083 and A considered as references for comparing. So their data were repeated again.
Q9: In the chapter Results, the authors cite a lot of references, it is not always clear that the results are taken from the literature or obtained by the authors.
Response: all results were extracted from our research but everywhere we use explanations of the other researchers we cited a reference.
Q10: Surface morphology in Figure 6 requires an explanation. Cracks on the surface raise doubts as to why mode D is better than the others.
Response: as mentioned in the manuscript, the cracks were also observed in the results of other researchers.
- Danaee, I.; Zamanizadeh, H.; Fallahi, M.; Lotfi, B. The effect of surface pre‐treatments on corrosion behavior of cerium‐based conversion coatings on Al 7075‐T6. Materials and Corrosion 2014, 65, 815-819.
- Fahrenholtz, W.G.; O'Keefe, M.J.; Zhou, H.; Grant, J. Characterization of cerium-based conversion coatings for corrosion protection of aluminum alloys. Surface and Coatings Technology 2002, 155, 208-213.
- Pinc, W.; Geng, S.; O’keefe, M.; Fahrenholtz, W.; O’keefe, T. Effects of acid and alkaline based surface preparations on spray deposited cerium based conversion coatings on Al 2024-T3. Applied Surface Science 2009, 255, 4061-4065.
- Hasannejad, H.; Aliofkhazraei, M.; Shanaghi, A.; Shahrabi, T.; Sabour, A. Nanostructural and electrochemical characteristics of cerium oxide thin films deposited on AA5083-H321 aluminum alloy substrates by dip immersion and sol–gel methods. Thin Solid Films 2009, 517, 4792-4799.
- Smith, G.N. Probability and statistics in civil engineering. Collins professional and technical books 1986, 244.
In the case of A, C and E coated plates, a local and non-uniform film has been formed but in the case of B and D, a cracked film has been observed. Cracks formed on the surface for the specimen order from the D regime are closer than that one formed on B. so this specimen could provide better protection.
However, cracks diminish the coating barrier effect, and as mentioned in the manuscript “The reason that the resistance of the coatings in the impedance test was low is due to the presence of cracks and these local deposits.”
Q11: It is necessary to enlarge Fig. 7.
Response; The size of the figure has been enlarged.
Q12: Why do the data in Fig. 4 and Fig. 9 not match for A?
Response; they are matched with each other but in Fig. 4 the axil value is different from that in Fig. 9. Fig. 9 was altered in the revised manuscript.
Q13: It is not clear table 7, what is the fundamental difference in ANN? How much is it necessary?
Response; As mentioned in the text, the performance and convergence of the ANN model are highly affected by the training parameters such as; the number of neurons and the training Algorithm. Thus, we seek to produce an appropriate structure for training datasets with neural networks. The changes in the number of neurons and training Algorithm resalted in different performance (as shown in Table 7 and Figure 10). The MAE value for LM training Algorithm is 14.1, while the MAE for BFG is 58.4. Therefore, an ANN model with better accuracy was achieved by changing the training Algorithm which demonstrated the necessity of this table.
Q14: Simulation is incomprehensible with what it is compared in Fig. 11
Response; Thanks for your comments, but that’s not correct. Please find the explanation for Fig. 11. As stated in the text, the accumulation of data points around the Y=X line displays that the anticipation of the experimental data is achieved with higher accuracy. The points near the regression line proved the efficiency of the developed model. For the ANFIS model, more data points were accumulated around the Y=X line. In addition, please check Table 8, the R2, MSE, and MAE values for three models were presented which the amount of R2 for ANN, ANFIS, and SVMR models were 0.99, 90, and 0.69, respectively. The value of MSE and MAE for the ANFIS model were 48.83 and 3.49 (lower value compared with ANN and SVMR). Based on Table 8, even though ANN and SVMR models displayed a reliable performance, it is explicit that the predictive potential of the ANFIS model is higher than other models.
Reviewer 3 Report
The present manuscript discusses the application of surface pretreatments (acid and/or basic combinations) to improve the cerium conversion coatings on AA5083 aluminum alloy and enhance the corrosion protection. The use of cerium conversion coatings on AA5083 is not new. There are several works on this subject like the ones I indicate below. A more exhaustive bibliographic search should be carried out and taken into account in the introduction and discussion of results. A simple search on "Google" has allowed to find these references:
- Method to improve corrosion resistance of AA 5083 by cerium based conversion coating and anodic polarisation in molybdate solution K Brunelli,M Magrini &M Dabalà Pages 223-232 Published online: 26 Nov 2013 https://doi.org/10.1179/1743278212Y.0000000001
- XPS and AES analyses of cerium conversion coatings generated on AA5083 by thermal activation J.M.Sánchez-AmayaaG.BlancobF.J.Garcia-GarciacdM.BethencourteF.J.Botanae https://doi.org/10.1016/j.surfcoat.2012.10.027
- Inhibition of 5083 aluminium alloy and galvanised steel by lanthanide salts M.AArenas M Bethencourt F.J Botana J de Damborenea M Marcos https://doi.org/10.1016/S0010-938X(00)00051-2
- Lanthanide salts as corrosion inhibitors for AA5083. Mechanism and efficiency of corrosion inhibition Yasakau, K.A., Zheludkevich, M.L., Ferreira, M.G.S. 2008 Journal of the Electrochemical Society 155(5), pp. C169-C177
It should be noted what novelty this work brings and consider these and other studies in the discussion of the results.
The corrosion protection provided by the cerium conversion coating is analyzed by cyclic voltammetry and EIS, and the morphology and composition is analyzed by SEM-EDS. The discussion is correct but should be improved taking into account previous studies.
In addition, computational techniques are used to predict the corrosion resistance of samples. The values ot Rct obtained by EIS and pHs used with etching and cleaning the samples are used to training three software. ANFIS is selected as the best software and the application time of conversion coating is introduced as another variable.
In the discussion it is indicated that the software allows to predict the Rct results of the samples but at no time is it validated. This is another weak points of the work. It would be necessary to use other pH values to validate the results obtained
Only 2 pH values 12 and 3 are used. Why are these pH values used? The results indicate that they are optimal, but no other values are used nor is their choice justified. It only indicates that the removal of the alumina layer is necessary to promote the growth of the conversion layer, and that the pH has to be higher than 11.5 or lower than 4 to dissolve the alumina layer.
pH values close to 14 or 0 could generate a very uneven surface and therefore improve the adhesion of the conversion layer even with shorter pretreatment times.
The software estimation is very hard to believe. The results indicate that the optimal values are 12 and 3. The software cannot give other values because they were only used to train it.
To say that the software is capable of predicting resistance values, other pH values (different from those used in training) must be used and verify that the real and predicted values match. If the validation is not carried out with other pH values, I consider that the part corresponding to modeling does not provide any information and may even be erroneous. Figure 12a shows resistance close to 200 KΩ using pH (1) = 10 and pH (2) = 3 but in the discussion it is indicated that alumina is stable at a pH lower than 11.5 and that it is necessary to remove it to generate good adhesion of the conversion layer. How can this result be justified?
Other coments:
The writing of the manuscript should be reviewed, below I indicate some examples
Line 38: The results demonstrated that the best deposition time was obtained for 10 min of deposition
Line 39: The coating resistance of cerium-based conversion coatings obtained
Line 59: They adsorb on the surface, react with aluminum and create fractures
Maybe: generate deffect, weaken, thinner the oxide layer
Line 500: The protection performance reduces.
Line 146: The condition used for the conversion coating application must be justified or referenced.
Line 165: potential range was from -1.2V to +0.8V with 1 mV/s scanning rate considering OCP
I don't understand what this means. Does the scan start at -1.2V or at the OCP?
lines 265-288 These paragraphs do not show the results, these first three paragraphs should go in the introduction or after indicating the results
Acronyms must be identified before first use
Figure 3: All samples must show the same range of potentials to be able to compare them properly “-1.2V to +0.8V” is indicated in experimental section
Voltammetries with aluminum alloys show great deviation in current densities due to the intermetallic particles present that act as defects in the passive layer.
How many replicas were made?
Line 303: oxygen limitation is visible, but this is not revealed in the coated panels:
What does this phrase mean?
Line 411: It is noticeable that H2O2 used in this work accelerates the rate of precipitations [60].
Why? Did you carry out any tests without using H2O2?
Figure 6: The cation in the figure is very confusing. The uncoated sample could be placed in position 6f.
Figure 7: The figure needs to be improved as nothing can be seen
Conclusions:
Line 595 and 597: Mean absolute error (MAE)
Scronyms must be defined before their first use, not twice in the end of the document
line 601: “Using the ANFIS model, the Rct of cerium-based conversion coatings was predicted with higher accuracy and precision”. This is not true, ANFIS was trained with the experimental data obtaining a good fit but It does not predict any value since it has not been validated
Author Response
Q1: A more exhaustive bibliographic search should be carried out and taken into account in the introduction and discussion of results.
- Method to improve corrosion resistance of AA 5083 by cerium based conversion coating and anodic polarisation in molybdate solution K Brunelli,M Magrini &M Dabalà Pages 223-232 Published online: 26 Nov 2013 https://doi.org/10.1179/1743278212Y.0000000001
- XPS and AES analyses of cerium conversion coatings generated on AA5083 by thermal activation J.M.Sánchez-AmayaaG.BlancobF.J.Garcia-GarciacdM.BethencourteF.J.Botanae https://doi.org/10.1016/j.surfcoat.2012.10.027
- Inhibition of 5083 aluminium alloy and galvanised steel by lanthanide salts M.AArenas M Bethencourt F.J Botana J de Damborenea M Marcos https://doi.org/10.1016/S0010-938X(00)00051-2
- Lanthanide salts as corrosion inhibitors for AA5083. Mechanism and efficiency of corrosion inhibition Yasakau, K.A., Zheludkevich, M.L., Ferreira, M.G.S. 2008 Journal of the Electrochemical Society 155(5), pp. C169-C177
Response:
The 1st article was cited in the manuscript in line 433;
The 2nd article was cited in the manuscript in line 152;
The 3rd and 4th article was cited in the manuscript in line 56.
Q2: It should be noted what novelty this work brings and consider these and other studies in the discussion of the results.
Response: effect of alkaline or acidic pretreatment on other alloys of Al had been studied but the effect of the sequence of alkaline and acidic conditions on AA5083 has not been investigated. Intermetallic in this alloy is different from the other types of AA, so we decided to study it. Additionally, the main purpose of this work was to find a correlation between electrochemical results and ANFIS modeling which no study has been done.
Q3: The corrosion protection provided by the cerium conversion coating is analyzed by cyclic voltammetry and EIS, and the morphology and composition is analyzed by SEM-EDS. The discussion is correct but should be improved taking into account previous studies.
Response: some declarations were added to the revised manuscript.
Q4: In addition, computational techniques are used to predict the corrosion resistance of samples. The values ot Rct obtained by EIS and pHs used with etching and cleaning the samples are used to training three software. ANFIS is selected as the best software and the application time of conversion coating is introduced as another variable. In the discussion it is indicated that the software allows to predict the Rct results of the samples but at no time is it validated. This is another weak point of the work. It would be necessary to use other pH values to validate the results obtained.
Response: Thanks for this comment. But, in this study, we investigated the difference between the acidic and alkaline conditions in order to find the suitable surface pretreatment for cerium conversion coating. Moreover, the nature of the acidic condition wouldn’t be very different from 3 to 5 and the nature in alkaline conditions wouldn’t be very different from 9 to 11. In order to prove the capability of ANFIS modeling for the prediction of Rct values, different values for pH of pretreatment-1 and pretreatment-2 were predicted by setting another input constant. Please find Figure 12 in the manuscript. Also, the error values and several criteria factors are evidence of the capability of the ANFIS model to simulate the experimental data set.
Q5: Only 2 pH values 12 and 3 are used. Why are these pH values used? The results indicate that they are optimal, but no other values are used nor is their choice justified. It only indicates that the removal of the alumina layer is necessary to promote the growth of the conversion layer, and that the pH has to be higher than 11.5 or lower than 4 to dissolve the alumina layer. pH values close to 14 or 0 could generate a very uneven surface and therefore improve the adhesion of the conversion layer even with shorter pretreatment times.
Response: As mentioned in Table 1, two surface pretreatment conditions were used to evaluate that which medium is more suitable for performing cerium conversion coating on aluminum alloy. Also, we have two pretreatment steps, and it was understood from the result that the sequence of this pretreatment will alter the chemistry and condition of the steel surface. In another word, alkaline pretreatment and then the acidic solution provided better condition compared with an acidic solution than alkaline pretreatment. Of course, at higher or lower pH values the deposition time may decrease, but it might result in more unfavorable conditions for steel surfaces in these unstable solutions. At lower pH values (lower than 3), the surface might become more destructed due to the presence of high hydrogen attach and harsh environmental operating conditions. For a higher pH value (higher than 12), the condition is the same as mentioned for acidic, which resulted in hash and dangerous conditions. It is worth noting that, in this study, we only focus on investigating the nature of the surface pretreatments and more extra experimental tests should be conducted to evaluate your statements (out of our goal).
Q6: The software estimation is very hard to believe. The results indicate that the optimal values are 12 and 3. The software cannot give other values because they were only used to train it. To say that the software is capable of predicting resistance values, other pH values (different from those used in training) must be used and verify that the real and predicted values match. If the validation is not carried out with other pH values, I consider that the part corresponding to modeling does not provide any information and may even be erroneous.
Response: That’s not correct, the models were validated in Table 9. As presented in this table, all three models satisfy the mentioned standard condition for each criterion which indicated the suitable performance and ability to predict. The error values and several criteria factors are evidence of the capability of the ANFIS model to simulate the experimental data set. In order to fulfill your question, please look at figure 12, which we added to the manuscript.
Q7: Figure 12a shows resistance close to 200 KΩ using pH (1) = 10 and pH (2) = 3 but in the discussion, it is indicated that alumina is stable at a pH lower than 11.5 and that it is necessary to remove it to generate good adhesion of the conversion layer. How can this result be justified?
Response: Based on the experimental and modeling results, by decreasing the PH (1) value (if pH (2) =3) the Rct decreases which demonstrated lower corrosion resistance of conversion coating. This is in agreement with experimental results.
Q8: Line 38: The results demonstrated that the best deposition time was obtained for 10 min of deposition.
Response: The statement was revised and highlighted in the text.
Q9: Line 39: The coating resistance of cerium-based conversion coatings obtained
Response: The statement was revised and highlighted in the text.
Q10: Line 59: They adsorb on the surface, react with aluminum and create fractures
Maybe: generate deffect, weaken, thinner the oxide layer
Response: According to the comment, the manuscript was revised.
Q11: Line 500: The protection performance reduces.
Response: the sentence was changed to “the protection performance of the coating reduces.
Q12: Line 146: The condition used for the conversion coating application must be justified or referenced.
Response: according to the comment the paragraph was revised and some sentences were added to the revised manuscript.
Q13: Line 165: potential range was from -1.2V to +0.8V with 1 mV/s scanning rate considering OCP
I don't understand what this means. Does the scan start at -1.2V or at the OCP?
Response: The sentences were edited in the manuscript. The scan rate was 1mV/s. the potential range should be altered. Scan rate has a great effect on the shape and accuracy of the cathodic and anodic Tafel branches and it should be selected intelligently. In this work, we applied 1mv/s just for comparison of the curves to obtain icorr and Ecorr.
Q14: lines 265-288 These paragraphs do not show the results, these first three paragraphs should go in the introduction or after indicating the results
Response: The manuscript was altered according to the comment. These paragraphs moved to the introduction section.
Q15: Acronyms must be identified before first use
Response: The manuscript was revised according to the comment and Acronyms were defined.
Q16: Figure 3: All samples must show the same range of potentials to be able to compare them properly “-1.2V to +0.8V” is indicated in experimental section
Voltammetries with aluminum alloys show great deviation in current densities due to the intermetallic particles present that act as defects in the passive layer.
How many replicas were made?
Response: 3 replications were made. As mentioned in Q13, the range of potential was altered. However, scan rate has a great effect on the polarization curves [ref]. because of different pretreatment applied on the AA5083 surface, so OCP of the samples is not similar and all of the test samples compared with the untreated AA5083 coupon. un-asymmetry of some curves may be related to the scan rate. It should be better to apply slower scan rates. anyway, for comparison scan rate of 1 mv/s could be valid.
Q17: Line 303: oxygen limitation is visible, but this is not revealed in the coated panels:
What does this phrase mean?
Response: In the case of untreated AA5083, oxygen limitation was observed but it was not observed for the other samples. This means that oxygen depletion occurred on the surface of the untreated AA5083 which is covered by a natural oxide layer.
Q18: Line 411: It is noticeable that H2O2 used in this work accelerates the rate of precipitations [60].
Why? Did you carry out any tests without using H2O2?
Response: No, we didn’t carry out any test without H2O2. According to the literature, researchers claimed that H2O2 could accelerate the rate of precipitation.
Q19: Figure 6: The cation in the figure is very confusing. The uncoated sample could be placed in position 6f.
Response: the caption is written according to the code of samples with different pretreatments.
Q20: Figure 7: The figure needs to be improved as nothing can be seen.
Response: The size of the figure has been enlarged.
Q21: Line 595 and 597: Mean absolute error (MAE)
Scronyms must be defined before their first use, not twice in the end of the document
Response: The statement was revised.
Q22: line 601: “Using the ANFIS model, the Rct of cerium-based conversion coatings was predicted with higher accuracy and precision”. This is not true, ANFIS was trained with the experimental data obtaining a good fit but it does not predict any value since it has not been validated
Response: As presented in Figure 12 and Table 9, the ANFIS model can predict all the other pH values, but the statement was revised based on your comment. The ANFIS model could simulate the Rct with high accuracy.
Round 2
Reviewer 2 Report
Authors answered on all comments
Author Response
Authors answered on all comments
Response: Thanks to the reviewer for his/her valuable points.
Reviewer 3 Report
Q1: A more exhaustive bibliographic search should be carried out.
The revision version only has modified introducing the four references, which I find in a quick search on “Google” on different sites of the test. The bibliobraphic search, is no my job, I don’t make a correct database search, so the references of cerium based conversion coating should be improved.
The 3rd and 4th articles appears in “line 58” to indicate the presence of intermetallic in AA5083. And first reference on line 435 to reference the effect of the H2O2
This articles and others should be used to indicate the state of the art of cerium conversion coating on AA5083 and compare the result with the present work
Q2: It should be noted what novelty this work brings and consider these and other studies in the discussion of the results.
“Response: effect of alkaline or acidic pretreatment on other alloys of Al had been studied but the effect of the sequence of alkaline and acidic conditions on AA5083 has not been investigated. Intermetallic in this alloy is different from the other types of AA, so we decided to study it. Additionally, the main purpose of this work was to find a correlation between electrochemical results and ANFIS modeling which no study has been done.”
It is true, but the state of the art of cerium conversion coating on AA5083 is not analysed in the introduction section, so it should be improved
Q3: the previous articles of cerium conversion coating were not used on the discussion of the results. The articles that I have indicated in the first review only are used to referenced the composition of the aluminium alloy and the effect of H2O2 in the etching. This should be improved
Q4: It would be necessary to use other pH values to validate the results obtained.
“In order to prove the capability of ANFIS modeling for the prediction of Rct values, different values for pH of pretreatment-1 and pretreatment-2 were predicted by setting another input constant. Please find Figure 12 in the manuscript.”
Ok, but ¿were the predicted Rct values validates with experimental test? In the experimental section it is indicated that pH 12 and 3 were used. However other pH values should be used
For example: acid pH 2, 3, or 4, and basic pH 9, 10, 11
The pH have a logarithmic relation with solubility of alumina, I think the software cannot predict Rct values just by inputting a single acidic pH value and a basic pH value
Q5: The table 1 only indicates that an acid solution and a basic solution are used. combined or not combined but "ONLY TWO PH VALUES ARE USED"
The inconsistency of the work is in the following. If only two pH values are used for training the software, how can the software predict the RCt value for treatments at other pHs?
What parameters are entered into the model to indicate the variation of the solubility of alumina with pH? These parameters should be clearly indicated in the text. Training with pH values 3 and 12 can assume that there is a linear trend between both values.
Another question that should be indicated in the text: Are the treatments in which STEP 2 does not apply are used in the training? (sample code A, B and C indicated in table 1).
If these are used, what pH(2) values are entered in training?
Q6: I consider that the part corresponding to modeling does not provide any information and may even be erroneous.
You indicated that the software can estimate the Rct values, so TO VALIDATE THE SOFTWARE a experimental test should be done at intermediate pH, for example pH(step 1)=4 and pH(step 2) =11 and compare the experimental value of Rct and predicted value by ANSIS and the desviation between both values should by indicated
Q13 and Q16: As you say, the scanning speed must be the same in all of them and it is well identified, 1 mV / s, but in the experimental part the range of potentials used must be adequately indicated. It is difficult to understand the difference of intervals present in figure 3.
It can be seen in figure 3 that:
AA5083 -1.1 to -0.5 (600 mV potential range)
A and D: -1.2 to -0.7 (500 mV potential range)
B and C: -1 to -0.3 (700 mV potential range)
E: -1.2 to -0.65 (potential range 550 mV)
All samples must have the same sweep, there are two options
Consider the OCP: for example start at OCP-300mV up to OCP + 500mV
Another option: fixed potential for all samples regardless of OCP, for example from -1 V to -0.3 vs calomel electrode
You can applied a current limit so that the sample does not corrode too much, but it does not seem that this criterion is applied in the tests since the scans that reach a higher potential are those with the greatest intensity of current.
It seems to me that different tests performed with different intervals are presented in a graph without having a common criterion.
The criteria used must be clearly indicated in the experimental part
Q18: It is noticeable that H2O2 used in this work accelerates the rate of precipitations [60]. Why? Did you carry out any tests without using H2O2? Response:
No, we didn't carry out any test without H2O2. According to the literature, researchers claimed that H2O2 could accelerate the rate of precipitation.
If you did not perform any test without using H2O2, the improvement that its addition produces cannot be seen in the results shown in the manuscript, so this sentence should be removed from the discussion. References should only be indicated in the experimental part. Line 182: "and the addition of 2ml H2O2 to increase the deposition rate of reaction [63,65]
You have used H2O2 since other authors indicate an improvement in deposition, but you did not carry out any tests
Author Response
Thank you again for reviewing and useful comments you sent. We tried our best to fulfill all the requested changes or additional description.
Q1: A more exhaustive bibliographic search should be carried out. The revision version only has modified introducing the four references, which I find in a quick search on “Google” on different sites of the test. The bibliobraphic search, is no my job, I don’t make a correct database search, so the references of cerium-based conversion coating should be improved. The 3rd and 4th articles appear in “line 58” to indicate the presence of intermetallic in AA5083. And first reference on line 435 to reference the effect of the H2O2. This articles and other should be used to indicate the state of the art of cerium conversion coating on AA5083 and compare the result with the present work
Response: The introduction part is improved in the revised version.
Q2: the state of the art of cerium conversion coating on AA5083 is not analysed in the introduction section, so it should be improved.
Response: In this regard, one paragraph is added to the introduction part.
Q3: the previous articles of cerium conversion coating were not used on the discussion of the results. The articles that I have indicated in the first review only are used to referenced the composition of the aluminium alloy and the effect of H2O2 in the etching. This should be improved.
Response: Some citation to the articles regarding the protection mechanism of CeCC applied on AA5083 is added to the discussion part of the revised version.
Q4: were the predicted Rct values validates with experimental test? In the experimental section it is indicated that pH 12 and 3 were used. However other pH values should be used. For example: acid pH 2, 3, or 4, and basic pH 9, 10, 11. The pH have a logarithmic relation with solubility of alumina, I think the software cannot predict Rct values just by inputting a single acidic pH value and a basic pH value
Response: Thanks for your comments. In this study, we trained the models based on the variable of experiments. Different surface pretreatment conditions were conducted to improve the corrosion resistance of cerium-based conversion coating on aluminum alloy and after that simulate the experimental variables with developed models. Investigation of the relation between pH with the solubility of alumina is out of our goals and may be considered in the future studied. Moreover, adding another independent variable into the model resulted in the complexity of the model which we do not suggest. The predicted Rct values are only suggested based on the reported error values. If we perform the experimental in another pH, what is the benefit of models?! The models were developed to reduce the experimental activities. As reported in the text we validate our constructed models in Table 9, in which several parameters were utilized to evaluate the ability of each model.
To sum up my explanation; according to the experimental tests of different surface pretreatment conditions at pH values of 3, 7, and 12 we trained three computational models and predict other pH values based on the reported errors. Of course, validation of predicted values with experimental results is helpful to understand the accuracy of models. But is almost impossible for us to get chances to run more tests on the subject in question. So, it would be highly appreciated to consider our predicted values as suggested values based on the reported errors and modeling results.
Q5: The table 1 only indicates that an acid solution and a basic solution are used. combined or not combined but "ONLY TWO PH VALUES ARE USED". The inconsistency of the work is in the following. If only two pH values are used for training the software, how can the software predict the RCt value for treatments at other pHs? What parameters are entered into the model to indicate the variation of the solubility of alumina with pH? These parameters should be clearly indicated in the text. Training with pH values 3 and 12 can assume that there is a linear trend between both values. Another question that should be indicated in the text: Are the treatments in which STEP 2 does not apply are used in the training? (Sample code A, B and C indicated in table 1). If these are used, what pH (2) values are entered in training?
Response: As you know, every computational model like ANFIS or neural network could find the nonlinear relationship between inputs and output parameters based on the trained dataset. When a model was trained with the dataset, it is ready to predict the output based on the inputs variables. In this study, we first reported the errors of each model and finally predict the Rct values at other pH values. The solubility of alumina with pH is not investigated and not considered in our models, because we set pH as an input variable and cannot use it as a dependent parameter. In the constructed models in which pretreatment-1 (acidic or alkaline cleaning: pH (1)), pretreatment-2 (acidic or alkaline cleaning: pH (2)), and deposition time in the cerium bath was set as the inputs for each model, and the charge transfer resistance of the coatings was selected as the output.
As mentioned, the models find the nonlinear relation between datasets in the specified domain of variables that were trained; for example in a model that was trained with pH=3 and pH=12, therefore all values between 3 and 12 could be predicted. We used pH values of 3, 7, and 12 for training the models. Moreover, the treatments in which STEP 2 does not apply are used in the training. We consider the pH (2) =7 for this situation. In another word, when step 1 or step2 has not been applied the pH value for that step was considered Neutral (pH=7). All in all, your comments were applied and mentioned in the text.
Q6: I consider that the part corresponding to modeling does not provide any information and may even be erroneous. You indicated that the software can estimate the Rct values, so TO VALIDATE THE SOFTWARE an experimental test should be done at intermediate pH, for example pH (step 1) =4 and pH (step 2) =11 and compare the experimental value of Rct and predicted value by ANSIS and the desviation between both values should by indicated
Response: With respect to your comment, I want to bring your attention to Table 8 and Table 9. As demonstrated, all models are valid with suitable accuracy. In order to validate and predict the exact value of output, we need the experimental data. But, in this study, we developed computational models to simulate the selected parameters. It is so interesting from your idea that focuses on issues that are not related to our goal and scope. I was wondering if considering this explanation as a final investigation because we cannot perform any more experimental tests. The main goal of this study is to evaluate the corrosion resistance of cerium-based conversion coating on aluminum alloy 5083 at different surface pretreatment conditions and develop computational models with suitable accuracy to simulate the corrosion resistance of coated Al-alloys at various pretreatment conditions. Of course, validation of predicted values with experimental results is helpful to understand the accuracy of models. But is almost impossible for us to get chances to run more tests on the subject in question. So, it would be highly appreciated to consider our predicted values as suggested values based on the reported errors and modeling results.
Q7: As you say, the scanning speed must be the same in all of them and it is well identified, 1 mV / s, but in the experimental part the range of potentials used must be adequately indicated. It is difficult to understand the difference of intervals present in figure 3. It can be seen in figure 3 that:
AA5083 -1.1 to -0.5 (600 mV potential range)
A and D: -1.2 to -0.7 (500 mV potential range)
B and C: -1 to -0.3 (700 mV potential range)
E: -1.2 to -0.65 (potential range 550 mV)
All samples must have the same sweep, there are two options. Consider the OCP: for example, start at OCP-300mV up to OCP + 500mV. Another option: fixed potential for all samples regardless of OCP, for example from -1 V to -0.3 vs calomel electrode. You can apply a current limit so that the sample does not corrode too much, but it does not seem that this criterion is applied in the tests since the scans that reach a higher potential are those with the greatest intensity of current. It seems to me that different tests performed with different intervals are presented in a graph without having a common criterion. The criteria used must be clearly indicated in the experimental part
Response: Yes, you are right. There are 2 ways for obtaining polarization curves. As we stated in the experimental part, the curves were gained in the potential range of +-300 mV around OCP, and as you know aluminum alloys don’t show the same curves due to different oxide film formations. On the other hand, if the curves were not symmetric may be as the result that the scan rate was not appropriate for it. But for comparison, we had to test at the same scanning rate. At this time, we have no access to the lab for tests again.
Q8: If you did not perform any test without using H2O2, the improvement that its addition produces cannot be seen in the results shown in the manuscript, so this sentence should be removed from the discussion. References should only be indicated in the experimental part. Line 182: "and the addition of 2ml H2O2 to increase the deposition rate of reaction [63,65]. You have used H2O2 since other authors indicate an improvement in deposition, but you did not carry out any tests
Response: according to the comment, the sentences were removed from the discussion part and moved to the experimental part.